# Impact of IBD-Associated Dysbiosis on Bacterial Quorum Sensing Mediated by Acyl-Homoserine Lactone in Human Gut Microbiota

**DOI:** 10.3390/ijms232315404

**Published:** 2022-12-06

**Authors:** Nathan Grellier, Marcelino T. Suzuki, Loic Brot, Alice M. S. Rodrigues, Lydie Humbert, Karine Escoubeyrou, Dominique Rainteau, Jean-Pierre Grill, Raphaël Lami, Philippe Seksik

**Affiliations:** 1Centre de Recherche Saint-Antoine, Assistance Publique-Hôpitaux de Paris, Hôpital Saint Antoine, Service de Gastroentérologie, Inserm, Sorbonne Université, F-75012 Paris, France; 2Laboratoire de Biodiversité et Biotechnologies Microbiennes, CNRS, Sorbonne Université, UAR3579, F-66650 Banyuls-sur-Mer, France; 3Observatoire Océanologique de Banyuls-sur-Mer, CNRS, Sorbonne Université, FR3724, F-66650 Banyuls-sur-Mer, France

**Keywords:** inflammatory bowel diseases, quorum sensing, acyl-homoserine lactones, gut dysbiosis, metagenomic, in silico

## Abstract

Intestinal dysbiosis is a key feature in the pathogenesis of inflammatory bowel disease (IBD). Acyl-homoserine lactones (AHL) are bacterial quorum-sensing metabolites that may play a role in the changes in host cells-gut microbiota interaction observed during IBD. The objective of our study was to investigate the presence and expression of AHL synthases and receptor genes in the human gut ecosystem during IBD. We used an in silico approach, applied to the Inflammatory Bowel Disease Multi’omics Database comprising bacterial metagenomic and metatranscriptomic data from stools of patients with Crohn’s disease (CD) (*n* = 50), ulcerative colitis (UC) (*n* = 27) and non-IBD controls (*n* = 26). No known putative AHL synthase gene was identified; however, several putative *luxR* receptors were observed. Regarding the expression of these receptor genes, the *luxR* gene from *Bacteroides dorei* was under-expressed in IBD patients (*p* = 0.02) compared to non-IBD patients, especially in CD patients (*p* = 0.02). In the dysbiosis situation, one *luxR* receptor gene from *Bacteroides fragilis* appeared to be over-expressed (*p* = 0.04) compared to that of non-dysbiotic patients. Targeting LuxR receptors of bacterial quorum sensing might represent a new approach to modulate the gut microbiota in IBD.

## 1. Introduction

The intestinal microbiota is defined as all microorganisms present in the digestive tract. Focusing on the bacterial compartment, it is estimated to comprise nearly 10^13^ bacterial cells throughout the digestive tract. They are distributed among three main phyla: Firmicutes, Bacteroidetes and Proteobacteria [1]. The gut microbiota has several physiological functions in humans, such as carbohydrate and lipid metabolism, a barrier function against pathogen colonisation and a role in the maturation of the host immune system [2,3,4]. When one or more of these functions are disrupted, a risk of developing intestinal diseases, such as inflammatory bowel diseases (IBD), arises. Even though several factors contribute to the development of IBD [5], it is now recognised that dysbiosis, defined as an imbalance in the composition and function of the gut microbiota, drives inflammation onset and perpetuation [6,7,8]. During IBD, a decrease in the Firmicutes phylum is observed, as well as an increase in Bacteroidetes and Proteobacteria, such as *Escherichia coli* (*E. coli*) [7]. These changes lead to alterations in bacterial metabolites, such as tryptophan and its derivatives, short chain fatty acid (SCFAs), bile acids and reactive oxygen species [9,10,11,12]. Besides these metabolites, our team has investigated a largely overlooked communication signalling mechanism used by gut bacteria, called quorum sensing (QS), which is known to modulate physiological responses in the three domains of life.

This widespread communication mechanism between bacteria is mediated by different type of signals. It is a density-dependent mechanism, allowing bacterial populations to coordinate gene expression and physiology [13]. In Gram-negative bacteria, there are multiple auto-inducing signals: auto-inducing signal type 1 (AI-1), type 2 (AI-2) and type 3 (AI-3). Among them, acyl-homoserine lactone (AHL)-driven QS (AI-1) is one of the most studied. AHLs are molecules composed of a lactone core and a carbon chain of variable length and saturation, produced by bacterial synthase proteins. The most common synthases are in the LuxI family, first described in *Aliivibrio fischeri* [14]. These amphiphilic molecules can passively diffuse into their environment and reach their cytosolic receptors in the LuxR family, in nearby bacterial cells. The binding of AHL to receptors acts as a transcription factor promoting the expression of virulence factors, bacterial growth, and biofilm formation [13]. This mechanism is used by *Pseudomonas aeruginosa* [15], but also by digestive pathogens, such as *Vibrio cholerae* and *Salmonella typhimurium* [16]. For commensal bacteria of the gastrointestinal tract, only three bacterial strains have been identified to have the couple LuxI/LuxR in healthy subjects using the Human Microbiome Project 1 database [17]. Nonetheless, bacteria in the Enterobacteriaceae family are known to have the LuxR-like receptor alone, called SdiA. These bacteria are not capable of AHL synthesis, but they can still sense them in their environment to express QS-mediated functions (a phenomenon described as eavesdropping) [18,19]. 

Our team demonstrated the presence of AHL in the human gastrointestinal tract, but also the differential presence of AHL between healthy subjects and IBD patients [20]. Healthy subjects had a higher concentration and diversity of AHL in their stool compared to IBD patients. A new AHL, the 3-oxo-C12:2 homoserine lactone (3-oxo-C12:2), was found to be over-represented in healthy subjects compared to IBD patients, especially in flare stages of the disease. Subsequently, this molecule showed in vitro anti-inflammatory effects in macrophage models in pro-inflammatory conditions, but also an activity in barrier function maintenance in the intestinal epithelium model [21,22,23]. These earlier results raise the question of the impact of IBD-associated dysbiosis on the AHL-mediated QS signal in the human gut ecosystem. 

To answer this question, we investigated the presence and expression of QS-related genes in non-IBD controls and IBD patients by an in silico approach using the open-access Inflammatory Bowel Disease Multi’omic Database (IBDMDB). To complement this approach, we also searched for AHL in stool samples of non-IBD controls and IBD patients of our Centre, using high-performance liquid chromatography coupled with mass spectrometry (HPLC-MS/MS).

## 2. Results

### 2.1. Presence of AHL in the Gut Microbiota of Controls and IBD Patients

To illustrate and confirm the presence of AHL in the gut ecosystem of IBD patients and non-IBD controls, HPLC-MS/MS detection was used to identify AHL spectral signatures. Faecal samples were obtained from ten healthy subjects with no history of digestive disease (non-IBD controls) and two relapsing Crohn’s disease (CD) patients. Detailed patients’ characteristics are available in the Appendix A.

As described by Cataldi et al. [24], identification of an AHL was defined by the presence of the precursor ion and the two possible product ions at the same retention time. HPLC-MS/MS detected an *m*/*z* that corresponds to a putative AHL molecule. Using one gram of lyophilised faeces per sample, thirteen different AHLs were detected. An internal standard was used to quantify AHL concentrations. The distribution of AHL among the samples is represented on a heat map (Figure 1).

In total, we observed 13 different AHLs in non-IBD controls group and 7 different AHLs in the IBD group. For example, subject 1 had all thirteen different AHLs in its faeces, whereas IBD patient B only had four. According to the non-IBD and IBD status, the diversity of AHL was similar to the previous study by Landman et al. [20]. Non-IBD controls had higher concentrations of AHL, especially for 3-oxo-C12:2 which had shown anti-inflammatory properties [21,22]. In summary, in this first study we confirmed the presence of AHL in human gut microbiota.

### 2.2. QS-Related Genes in Human Gut Microbiota

#### 2.2.1. Patients’ Characteristics of the in Silico Study

To search for QS-related genes in gut microbiota, we used data from the Inflammatory Bowel Disease Multi’omics Database [25]. This population of 132 subjects had undergone sampling of faeces every 2 weeks for 52 weeks according to a defined protocol. Among the 132 subjects of the total cohort, 103 were included in the present work, since these subjects underwent paired microbiota metagenomic and metatranscriptomic sequencing at the same time point (albeit that could vary between subjects during the follow-up). We analysed one faecal sample for each subject. There were 26 non-IBD controls and 77 IBD subjects (50 CD and 27 UC). The demographic and clinical data are shown in Table 1. Detailed metadata are available in the Appendix A.

Most of the patients were in clinical remission. In the CD group, according to the Harvey–Bradshaw Index (HBI), 34 patients (68%) had a non-active disease (HBI < 4), 11 patients (22%) had a mild activity (HBI ≤ 4–8 ≤), only 1 patient (2%) had a moderate activity (HBI < 8–12 ≤), and none had a severe activity (HBI > 12). Four CD patients had an unknown HBI at sampling [26]. In the UC group, according to the Simple Clinical Colitis Activity Index (SCCI), 19 patients (73%) had a non-active disease (SCCAI 0–2 ≤), 7 patients (27%) had a mild activity (≤ 3–5 ≤), there were no moderate or severe diseases, and one patient had unknown SCCAI at sampling [27]. The biological activity (C-reactive protein) and the endoscopic activity of the disease were unknown at the time of sampling. 

#### 2.2.2. Microbiota Composition According to Disease Phenotypes

The microbiota composition and the diversity distribution among the three groups (non-IBD controls, CD, and UC patients) were assessed using open-access taxonomic profiles [25]. The diversity and microbiota composition of each group are shown in Figure 2.

Regarding alpha diversity, CD patients had a lower richness in terms of the number of observed species compared to non-IBD patients (*p* = 0.025) and UC patients (*p* = 0.020). The Shannon index was not significantly different between the three groups (*p* = 0.21). Concerning beta diversity according to the disease phenotype, no significant difference was shown between the three groups based on a Permanova analysis (*p* = 0.49). At the phylum level, the Verrucomicrobia phylum was over-represented in IBD patients (*p* = 0.006), but there were no significant differences for Firmicutes (*p* = 0.97), Bacteroidetes (*p* = 0.79) and Proteobacteria (*p* = 0.24). At the species level, *Akkermansia muciniphila* showed a higher relative abundance in CD patients (*p* = 0.004), and *Alistipes putredinis* was over-represented in non-IBD controls (*p* = 0.04). There were no significant differences found for other species. Detailed analyses are available in Appendix A. In conclusion, non-IBD controls and IBD patients were not strikingly different in terms of microbiota composition and diversity.

#### 2.2.3. QS-Related Genes

In the present study, we searched for QS-related genes comprising AHL-synthases and AHL-receptors [28]. We established a catalogue of putative genes. The synthase genes used were *luxI*-homologues described in Doberva et al. [29], exhibiting 603 synthase genes. Ten *luxR* homologous receptor genes were used [17,30,31,32,33,34]. All gene sequences and accession numbers are listed in Appendix A. Appendix A summarise the genes and their functions. 

#### 2.2.4. Presence of Homologous QS-Related Genes

To improve our chances of showing the presence of QS-related genes, we used the Basic Local Alignment Search Tool (BLAST) [35] in *tblastn* mode with an expected value (e-value) < 10^−20^ and without identity restriction. No synthase gene was found in the assembled gene files of our samples. Concerning receptors genes, *sdiA* homologous genes were found in seven patients, five in CD patients, and two in non-IBD controls. The mean percentage identity to *sdiA* from *E. coli* was 90.7% ± 4.7 SEM. We found other *luxR*-like genes close to homologues from *Bacteroides fragilis* (*B. fragilis*) or *Bacteroides dorei (B. dorei)* in all samples. The detailed number of hits are available in Appendix A.

#### 2.2.5. AHL Receptor Genes According to Disease Phenotypes

Beyond the presence of LuxR receptors homologues, we used *blastn* against raw reads from the metagenome to quantify the relative abundance of each gene and against raw reads from the metatranscriptome to quantify their expression (Figure 3).

No alignment for synthase genes was found using *blastn* against raw reads with default BLAST parameters (all identity percentage, all expected values), meaning that our reference genes were not present in our samples. Regarding receptor genes, we found *luxR* gene reads in all samples. However, there was no significant difference in terms of relative abundance in the metagenome among non-IBD controls, CD, and UC patients. The receptor gene *luxR1* of *B. dorei* was under-expressed in IBD patients compared to non-IBD patients (*p* = 0.023). It was under-expressed in CD patients compared to non-IBD (*p* = 0.023), but not in UC patients compared to non-IBD patients (*p* = 0.11). The detailed number of hits are shown in Appendix A, and analysis of each group are shown in Appendix A.

#### 2.2.6. QS-Related Genes According to IBD-Associated Dysbiosis Status

As already observed [25], dysbiosis is not systematically correlated to IBD. To evaluate our results relative to dysbiosis status we focused our analysis on IBD-associated dysbiosis status. To define it, we used the linear discriminant analysis effect size (LEfSe) method with rarefied data to identify discriminating bacterial species between non-IBD controls and IBD patients [36]. Four bacterial species were identified and used to define dysbiosis status (Figure 4 and see Section 4). Two samples were excluded from analysis because of their abnormally high relative abundance of *Akkermansia muciniphila* (outliers). There were 47 samples (4 non-IBD, 24 CD and 17 UC) in the dysbiotic group, and 54 samples in the non-dysbiotic group. The alpha and beta diversity of these two groups are shown in Figure 4.

Here, we observed a lower number of species and a lower Shannon index in the dysbiotic group (*p* = 0.003 and *p* = 0.014 respectively). Beta diversity was significantly different between the two groups using a one-way Permanova analysis (*p* = 0.034). The same *blastn*-based method above was used to quantify relative abundance and expression of *luxR*-like genes. Different groups and subgroups were used: ND (non-dysbiotic) (*n* = 54); D (dysbiotic) (*n* = 47); ND-non-IBD (non-dysbiotic non-IBD) (*n* = 20); ND-IBD (non-dysbiotic IBD) (*n* = 34); and D-IBD (dysbiotic IBD) (*n* = 41).

No significant difference was found in terms of relative abundance of all tested genes. Regarding expression, only *luxR4* from *B. fragilis* [34] appeared to be over-expressed in the dysbiotic group compared to the non-dysbiotic group (*p* = 0.038) (Figure 5). This result was confirmed in sub-group analysis: dysbiotic-IBD patients (D-IBD) compared to all non-dysbiotic subjects (ND) (*p* = 0.043) and dysbiotic IBD patients (D-IBD) compared to non-dysbiotic IBD (ND-IBD) subjects (*p* = 0.035). There was no significant difference concerning expression of the *luxR1* of *B. dorei*. Detailed analyses for each gene and groups are available in Appendix A.

## 3. Discussion

Information about the involvement of AHL-mediated QS in the balance of the human gut ecosystem is scarce [37]. Our team previously highlighted the role of AHL as metabolites of interest in inflammation pathways. Besides the confirmation of AHL presence in human gut lumen, the results presented here showed that the expression of *luxR* from *Bacteroides* spp. could be linked to IBD-associated dysbiosis. These results were obtained from an in silico approach, showing potential as a new target for microbiota modulation in IBD (Figure 6).

We were unable to identify synthase genes, neither when using flexible search criteria with protein sequences nor a stricter search with nucleotide sequences [17,29,35]. These results raise the fundamental question of the presence of AHL synthase genes in the human intestinal microbiota. Based on our metabolomic results, synthase genes should have been found in the human gut. The first hypothesis is that synthase genes used for alignment might be completely different from those potentially present in the human gut. Referring to previous studies using the same method [17], the identified bacteria carrying the LuxI/LuxR pair genes *(Hafnia alvei* ATCC 51873, *Edwardsiella tarda* ATCC 23685, and *Ralstonia* sp. 5_7_47FAA) are not common bacteria in the intestinal ecosystem. In the present work, the synthase genes used as query were mostly found in marine ecosystems (*Vibrio* spp. or *Aeromonas* spp.) and in pathogenic bacteria, such as *Pseudomonas aeruginosa* or *Citrobacter rodentium* (Appendix A), which could have missed genes in the gut microbiota. Another more likely explanation would be a sequencing depth insufficient to include rarer AHL producing bacteria. If that is the case, those genes would be rarely represented in the metagenome, and the potential prominent role of AHL in human gut physiology would be questioned. Beside the absence of synthase genes, the presence of receptor genes might be viewed as a proxy of AHL presence in the human gut. Indeed, if LuxR receptors are present in the gut ecosystem and considering AHL as their main activating signal, this indicates that LuxR receptors may be an indirect witness of the presence of AHL. Thus, it appears crucial to use other methods to discover the AHL synthase genes. For example, using wet lab experiments and challenging bacterial communities with a high amount of AHL might induce over-expression of synthase genes (auto-inducing signal) and reshape the gut microbiota, as has been shown for other QS signals [38].

There are several limitations to our study. First, most of the selected patients were in clinical remission. This issue has been reported by IBDMDB investigators, but the cohort was not originally designed to be stratified by disease activity [25]. Secondly, the controls selected as non-IBD were patients who consulted for digestive symptoms, for whom the digestive endoscopy and pathological findings were not in favour of IBD. In other terms, there were not strictly healthy subjects. This may explain the small difference in microbiota diversity between non-IBD and IBD patients. Finally, we selected 103 patients, with metagenomic and metatranscriptomic sequencing on the same sample as inclusion criteria. This condition forced us to choose samples at different timelines of the cohort rather than at the beginning, skewing our results towards low activity in IBD patients (treatment exposition) and small differences in microbiota composition. These three factors, i.e., mild activity diseases, non-strict healthy subjects, and our inclusion criteria, motivated us to analyse our results considering dysbiosis, not only disease phenotypes. We based our dysbiosis definition on differential bacterial species representation between our control group and IBD patients [36], allowing us to differentiate two groups. The richness and the diversity of our dysbiosis group were lower compared to the non-dysbiosis group. This was in accordance with literature on IBD-associated dysbiosis, confirming our choice [39]. From this definition, we obtained interesting differential expression of *luxR* genes from *B. fragilis*. 

Regarding receptor genes, the *luxR* receptor genes from *Bacteroides* spp. were shown to be involved in biofilm formation, bacterial growth, and the acquisition of antibiotic resistance when subjected to the C6-HSL [34]. In our study, one of these receptor genes was differentially expressed in the dysbiosis and non-dysbiosis groups. When we look closely at the structure of LuxR family proteins, they exhibit two functional domains: an amino-terminal auto-inducing binding domain and a carboxy-terminal DNA-binding domain [28]. LuxR1 *B. dorei* and LuxR4 *B. fragilis* are a putative transcriptional regulator GerE and a putative DNA-binding response regulator, respectively (See Section 4). Both protein sequences do not contain auto-inducing binding regions. In fact, the genes used in our work correspond to the protein regions that bind bacterial DNA and act as transcription factors. Note that Pumbwe et al. showed their over-representation in *B. fragilis* conditioned with an AHL-supplemented medium [34]. This leads us to hypothesise that another mechanism could activate or repress LuxR proteins without the auto-inducing domain in the presence of AHL. In other bacterial species, for example in *Vibrio harveyi*, the LuxR protein can be involved in other QS systems through quorum regulatory small RNAs (Qrr) that repress LuxR protein [28]. Those Qrr are dependent on other auto-inducing signals, such as type 2 auto-inducing signal (AI-2). We can hypothesise that LuxR protein, containing only DNA-binding, can also be activated through a non-canonical AHL signalling pathway. Mechanistic in vitro studies could be performed to explore the interaction between these LuxR, AHL and other QS signals. Concerning *Bacteroides* spp., enterotoxigenic *B. fragilis* has been reported to be associated with IBD, but there is no evidence for *B. dorei* involvement in the pathophysiology of IBD [40,41]. Our approach did not allow us to differentiate enterotoxigenic from non-toxigenic *B. fragilis* strains even though enterotoxigenic strains are usually associated with IBD gut microbiota. We do believe that the dynamic of bacterial communities may be influenced by LuxR transcriptional factors; however, the lack of work regarding their role in dysbiosis encourage us to explore this idea. Figure 6 summarises the impact of IBD-associated dysbiosis on LuxR receptors. 

In conclusion, our current findings provided novel insights into AHL-driven QS in IBD gut microbiota, showing that the expression of LuxR receptors in *B. dorei* and *B. fragilis* are modulated by dysbiosis. Nevertheless, mechanisms involving the response of LuxR family proteins must be explored to understand their implication in IBD-associated dysbiosis. 

## 4. Materials and Methods

### 4.1. HPLC-MS/MS Samples Pre-Treatment and AHL Extraction

Samples of IBD patients originated from the gastroenterology department of Saint-Antoine Hospital, Paris, France. They were collected in September 2022. All patients gave written consent for stool donation as part of the Suivitheque study (Institutional Review Board 00003835). The stool was frozen at −80 °C. Each sample was lyophilised from the whole stool sample for at least 48 h and homogenised. One gram of lyophilised faeces was used for each sample extraction [20]. The detailed AHL extraction protocol is available in Appendix A. Two microlitres of the internal standard N-hexanoyl-L-homoserine lactone-d3 (C6d3-HSL, Cayman Chemical) at a concentration of 2.5 mM per tube was used as an intern standard. For mass spectrometry, extracted samples were taken up in 1 ml of acetonitrile and then injected at 2 μL.

### 4.2. AHL Detection with HPLC-MS/MS

Samples were analysed using an LC-20ADXR chromatographic system (Shimadzu, Kyoto, Japan) in tandem with a QTRAP 5500 quadrupole linear ion trap MS/MS spectrometer system (SCIEX, Vaughan, ON, Canada). MS detection was performed using electrospray ionisation (ESI) in positive mode using the Multiple Reaction Monitoring (MRM) function of the analyser. Spectral data acquisitions were processed using Analyst software (v1.6.3) in the multiple reaction monitoring mode. As described by Cataldi et al., identification of an AHL (designated by the *m*/*z* of the precursor ion [M+H]+) was defined by the simultaneous presence of the precursor ion and the two possible product ions ([M+H −101]+, neutral loss and 102 or lactone moiety) [24]. The AHLs were expressed in relative concentrations and normalised relative to an internal standard (C6-d3-HSL). The quantifications were performed with Multiquant software (v3.0.2) (SCIEX, ON, Canada).

### 4.3. Inflammatory Bowel Disease Multi’omics Database and Samples

The assembled gene files, raw data, and taxonomic profiles used were freely available at https://ibdmdb.org/tunnel/public/summary.html (accessed on 28 September 2022) [25]. Relative abundances of taxonomic profiles were parsed using bash, *awk*, and *sed* to parse different taxonomic ranks. Relative abundance values were multiplied by the total reads number of each sample to approximate the absolute number of each species. Qiime2 was used to rarify the bacterial species in each sample to 50,000 sequences per sample, and calculate the alpha-diversity indices and beta-diversity values [42]. Rarefied bacterial species data are available in Appendix A.

### 4.4. Dysbiosis Definitions

The definition of dysbiosis was determined using the LEfSe method with bacterial species on rarefied data [36]. Species with an LDA score of >2 were considered discriminating (*p* < 0.05). Among the 103 patients, 2 paediatric CD patients had a high relative abundance of *Akkermansia muciniphila*: HSMA33NQ at 75.9% and HSM5MD5P at 8.5%. It has been described that the abundance of this species was decreased in paediatric CD patients. In order to not distort our definition of dysbiosis, these two patients were excluded from the LEfSe analysis [43]. All dysbiosis analyses were performed on 101 patients. Once the four bacteria were identified, the specificity thresholds were determined with a receiver operating characteristic curve (ROC curve), and the most specific species for each condition were used. Samples had to have at least 3 of the 4 discriminating bacterial species to be classified as dysbiotic. ROC curves are available in Appendix A. Dysbiotic samples list is available in Appendix A.

### 4.5. QS-Related Genes

The numbers used to distinguish the different *luxR* genes of *B. fragilis* (*luxR1*, *luxR2*, *luxR3*, *luxR4*) were determined arbitrarily. *luxR1 B. dorei* is a homologous gene of *luxR1 B. fragilis* with a percentage identity of 94%. Table 2 shows *luxR* numbers and their corresponding accession numbers. 

### 4.6. Sequences Alignment

We used the Basic Local Alignment Search Tool (BLAST) [35] in *tblastn* mode on assembled gene files using protein sequences to assess the presence or absence of the gene of interest. We used *blastn* mode to quantify genes reads using nucleotide sequences. The parameters in *tblastn* were an e-value of < 10^20^, with all identity percentages (ID%). Queries on the metagenome raw files and metatranscriptome raw files were performed using nucleotide sequences of our reference QS-related genes in *blastn*. The *blastn* parameters were an e-value of < 10^30^, with variable ID%. For *luxR1 B. fragilis*, the ID% was >80%, to be specific to *B. fragilis*. For *luxR2* to *luxR4 B. fragilis*, all ID% values were used because *B. fragilis* is the only bacterial species to have these genes in the NCBI GenBank. For *luxR1 B. dorei*, we used a threshold higher than 97% to be specific to *B. dorei*. The relative abundance of the genes was expressed with the number of gene reads normalised by the total reads number in the sample. Expression of QS-related genes was expressed by the number of gene reads in the metatranscriptome normalised by the total reads number in a sample over the relative abundance of the gene. Samples with no gene reads in the metagenome were not used for the analysis of gene expression. 

### 4.7. Statistical Analysis

GraphPad Prism V.8.0.1 (San Diego, CA, USA) was used for all analyses and graph preparation. For all graph data, the results are expressed as mean ± SEM, and statistical analyses were performed using the two-tailed non-parametric Mann Whitney U test or Kruskal–Wallis test with Dunn’s multiple comparison test. Qiime2 was used to rarify to 50,000 sequences per sample the bacterial species in each sample and calculate the alpha-diversity indices and beta-diversity [42]. The number of observed species and the Shannon diversity indices were calculated using rarefied data (depth = 50,000 reads/sample), which are used to characterise species diversity in a community. Statistical significance of sample grouping for beta diversity analysis was performed using Permanova method (9999 permutations) with Bray-Curtis dissimilarities. Paleontological Statistic Package (PAST) V4.03 software was used for the beta-diversity analyses [44]. Differences with a *p* value < 0.05 were considered significant. Differential analysis was performed using the linear discriminant analysis effect size (LEfSe) pipeline [36]. 

## Figures and Tables

**Figure 1 ijms-23-15404-f001:**
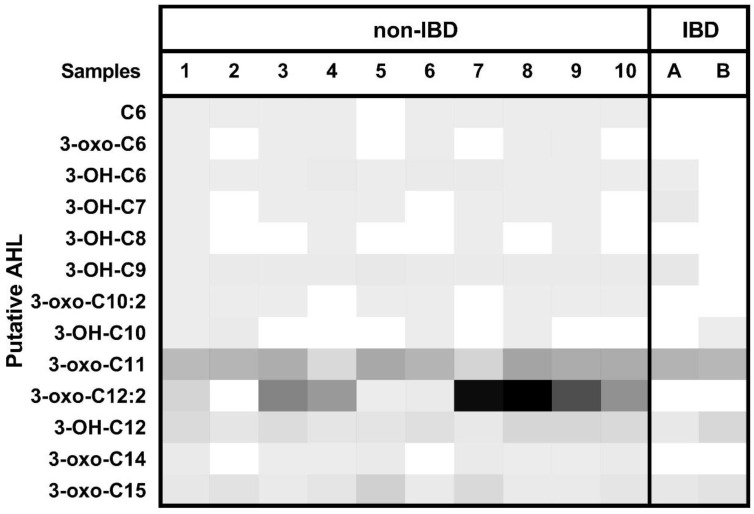
Heatmap of putative AHL profiles in non-IBD controls and IBD patients. The grey color bar intensity indicates the AHL *m*/*z* integrated peak area from zero (white) to the highest concentration (black). IBD: inflammatory bowel disease patients. Numbers and letters represent sample names.

**Figure 2 ijms-23-15404-f002:**
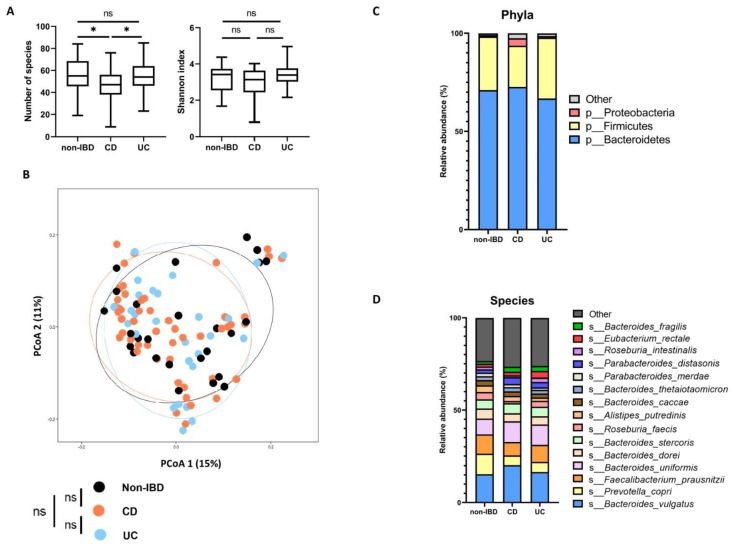
Microbiota composition according to disease phenotype. (**A**) Number of observed species and Shannon index describing alpha diversity of bacterial microbiota (Kruskal–Wallis test with Dunn’s multiple comparison test). (**B**) Principal coordinate analysis of Bray–Curtis distances with each sample coloured according to the disease phenotype. PCoA1 and PCoA2 represent the top two principal coordinates that captured most of the distance. The fraction of distance captured by the coordinate is given as a percentage. Groups were compared using the Permanova method. (**C**,**D**) Global composition of bacterial microbiota at the phyla and species level. Non-inflammatory bowel disease controls (non-IBD) and patient subgroups are labelled on the *x*-axis and expressed as the relative abundance in the metagenome for each group. In all panels, ns: non-significant, * *p* < 0.05, CD: Crohn’s disease, UC: ulcerative colitis.

**Figure 3 ijms-23-15404-f003:**
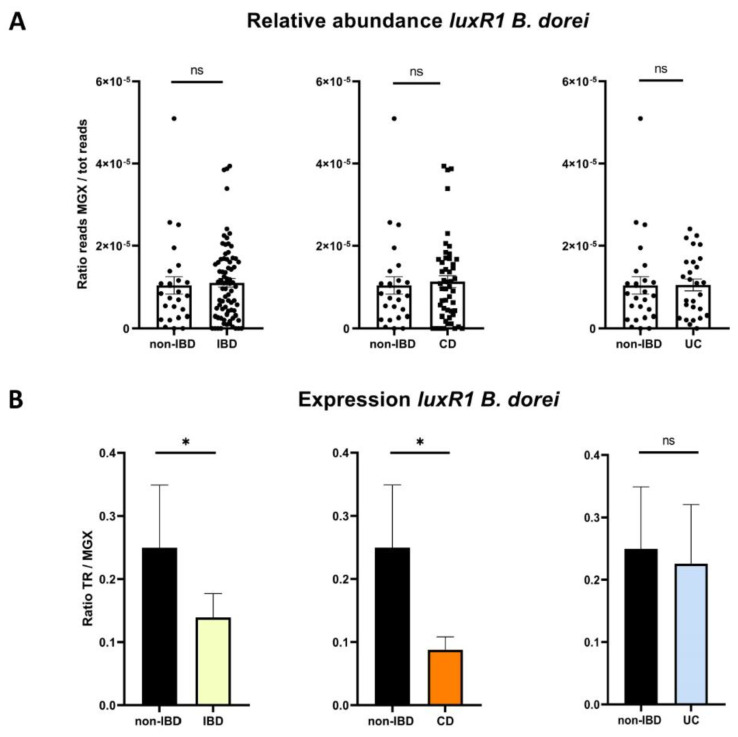
Receptor gene *luxR1 Bacteroides dorei (B. dorei)* according to disease phenotype. (**A**) Relative abundance of *luxR1 B. dorei* expressed in the ratio of gene reads recovered by *blastn* searches over the total number of reads in the sample (Mann–Whitney U test). (**B**) Expression of *luxR1 B. dorei* expressed by the ratio of gene reads recovered by *blastn* searches in the transcriptome over the gene reads in the metagenome (Mann–Whitney U test). In all panels, ns: non-significant, * *p* < 0.05, CD: Crohn’s disease, UC: ulcerative colitis, non-IBD: controls, MGX: metagenome, TR: transcriptome.

**Figure 4 ijms-23-15404-f004:**
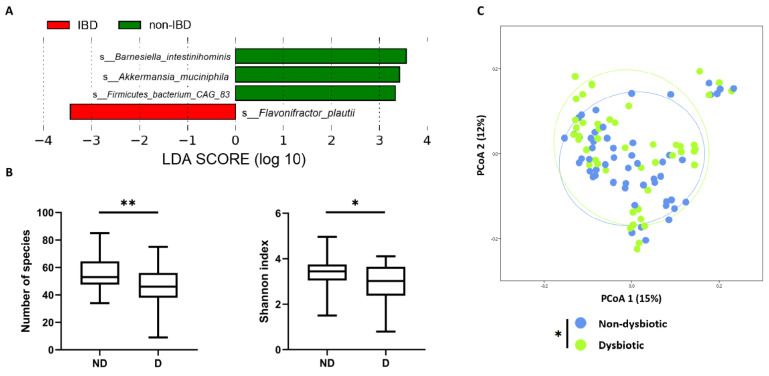
Discriminating bacterial species between non-IBD controls and IBD patients and diversity according to dysbiosis. (**A**) LEfSe of the four main discriminating species between IBD patients and non-IBD controls. (**B**) Alpha diversity indexes according to dysbiosis (Mann–Whitney U test). (**C**) Principal coordinate analysis of Bray–Curtis distance with each sample coloured according to the disease phenotype. PCoA1 and PCoA2 represent the top two principal coordinates that captured most of the distance. The fraction of distance captured by the coordinate is given as a percentage. Groups were compared using Permanova analysis. In all panels, * *p* < 0.05, ** *p* < 0.005. ND: non-dysbiotic, D: dysbiotic.

**Figure 5 ijms-23-15404-f005:**
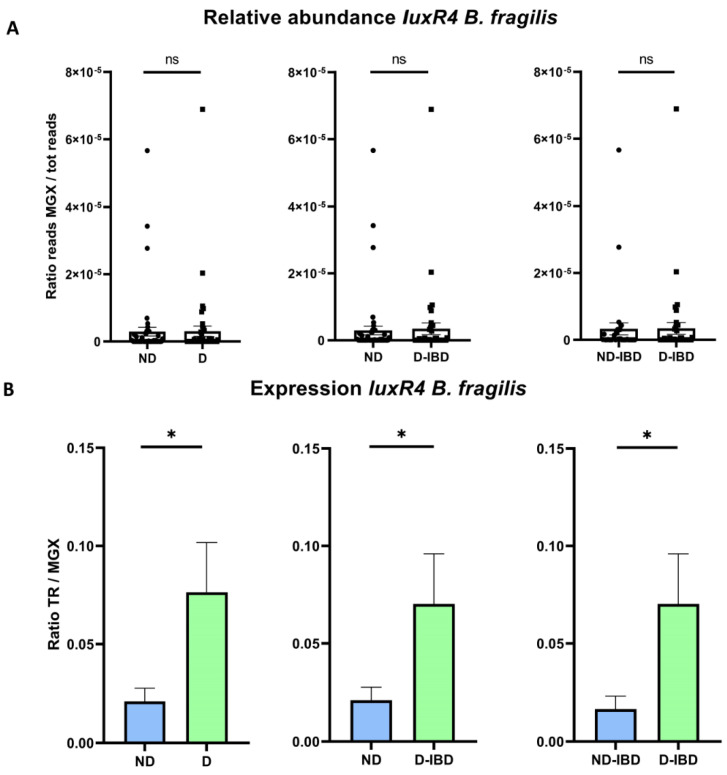
Relative abundance and expression of *luxR4 Bacteroides fragilis (B. fragilis)* according to dysbiosis status. (**A**) Relative abundance of *luxR4 B. fragilis* expressed as the ratio of gene reads over the total number of reads in the sample (Mann–Whitney U test). (**B**) Expression of *luxR4 B. fragilis* expressed as a ratio of gene reads in the transcriptome over the gene reads in the metagenome (Mann–Whitney U test). In all panels, ns: non-significant; * *p* < 0.05, ND: non-dysbiotic, D: dysbiotic, D-IBD: dysbiotic IBD, ND-IBD: non-dysbiotic IBD, MGX: metagenome, TR: transcriptome.

**Figure 6 ijms-23-15404-f006:**
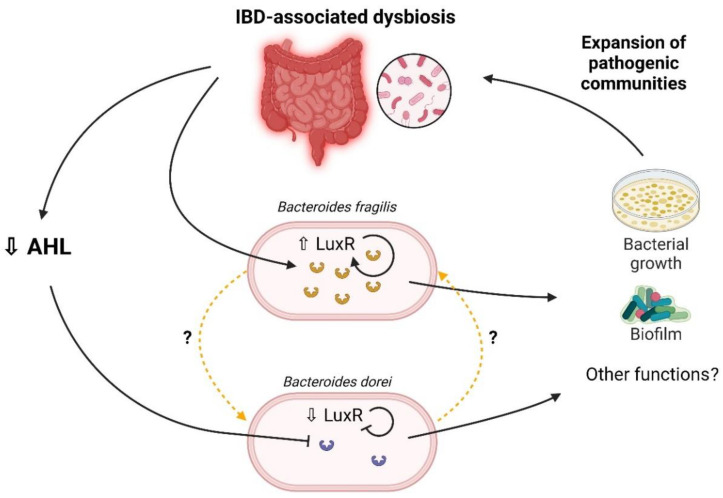
Impact of IBD-associated dysbiosis on LuxR receptors. Created with https://BioRender.com (accessed on 26 October 2022).

**Table 1 ijms-23-15404-t001:** Patients’ characteristics. IBD: Inflammatory bowel disease; CD: Crohn’s disease; UC: Ulcerative Colitis; F: Female, M: Male; SEM: Standard error of the mean. * Paediatric: patients < 18 years old; ** New diagnosis: diagnosis of IBD at inclusion in the cohort; Antibiotics: exposition to antibiotics 2 weeks before sampling; Steroids: exposition to steroids 2 weeks before sampling; HBI: Harvey–Bradshaw index score; SCCAI: Simple Clinical Colitis Activity index.

	Non-IBD *n* = 26	CD *n* = 50	UC *n* = 27
Sex (F/M)	11/15	23/27	17/10
Age (years) (mean +/− SEM)	29.7 ± 4.0	25.9 ± 2.4	29.2 ± 3.5
Paediatric * (*n*)	12	26	12
Disease duration (years) (mean +/− SEM)		6.0 ± 2.8	5.7 ± 1.4
New diagnosis ** (*n*)		35	14
Antibiotics	0	8	1
Steroids	0	13	2
History of surgery	0	10	0
Clinical activity score (mean +/− SEM)		HBI: 2.1 ± 0.3 Unknown: 4 patients	SCCAI: 1.6 ± 0.2 Unknown: 1 patient

**Table 2 ijms-23-15404-t002:** Gene list from Bacteroides fragilis and Bacteroides dorei.

	Accession Number	Function
*luxR1 B. fragilis*	BF9343_2602	Putative DNA-binding response regulator
*luxR2 B. fragilis*	BF9343_2858	Putative sigma factor
*luxR3 B. fragilis*	BF9343_3797	Putative LuxR-family regulatory protein
*luxR4 B. fragilis*	BF9343_4003	Putative transcriptional regulator GerE
*luxR1 B. dorei*	NZ_LR699004.1:3959839-3960450	Putative DNA-binding response regulator

## Data Availability

The data presented in this study are available on request from the corresponding author.

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
