# Peer review of "Impact of IBD-Associated Dysbiosis on Bacterial Quorum Sensing Mediated by Acyl-Homoserine Lactone in Human Gut Microbiota"

_ijms, 2022, doi:10.3390/ijms232315404_

Round 1
Reviewer 1 Report
The manuscript entitled " Impact of IBD-associated dysbiosis on bacterial quorum sens-2 ing mediated by acyl-homoserine lactone in human gut micro-3 biota" by Grellier N. and coworkers aims at investigating how bacterial quorum sensing is affected by dysbiosis in patients suffering from IBD. Authors focused first their investigation on the quantification in fecal samples of acyl-homoserine lactone by HPLC-MS/MS (n=10 non-IBD and n=2 IBD). Despite the limitations due to the small number of samples, Authors found higher concentration of AHL in non-IBD in comparison to IBD patients and notably the 3-oxo-C12:2 compound which have been previously identified for its anti-inflammatory properties.
Authors also explored the presence of both putative synthase and receptor genes responsible for AHL production and sensing in metagenomic and metatranscriptomic dataset in a cohort of individuals made of control, UC and CD (n=26 non-IBD, n=50 CD and n=27 UC). Whereas no significant results (except at the richness level) are found at the gut microbiome compositional level, Authors point to an overexpression of luxR1 gene of B. dorei in CD. By clustering the individuals according to the dysbiosis status of their microbiome they have also explored how microbial communication is affected in such condition. Their findings highlight an increased expression in dysbiotic microbiome setting of luxR4 gene expression from B. fragilis.
On the overall, the paper adds new pieces of data related to potential dysfunction in the gut microbial dialogue during dysbiosis. It also points to QS functions that may be used as potential target to improve the quality of life of IBD patients suffering from microbiome dysbiosis.
Author Response
We thank reviewer 1 for his/ her complete and detailed analysis of our manuscript. We share his/ her view of our work. We fully agree with his/ her conclusions.
Reviewer 2 Report
The manuscript “Impact of IBD-associated dysbiosis on bacterial quorum sensing mediated by acyl-homoserine lactone in human gut microbiota” looks interesting. But I suggest revision of the followings
1. I suggest to use better term for dialogue in “Acyl-homoserine lactones (AHL) are bacterial quorum sensing metabolites that may play a 16 role in the changes of the dialogue between host and gut microbiota observed during IBD”
2. In the abstract “In dysbiosis situation, one luxR receptor gene from Bacteroides fragilis appeared to be over-expressed (p=0.04) compared to non-dysbiotic patients” but how did you differentiate toxic and non-toxigenic strains bacteria? Since only toxigenic bacteria are linked to IBD or CD
3. Why only AHLs? What about other signalling molecules? The Gram-positive bacteria produces multiple QS peptides
4. Overall, this hypothesis or concept apply only for few bacteria. What about other dysbiosis bacteria?
5. What about ethical clearance of the samples “Samples of IBD patients were originated from the gastroenterology department of 347 Saint-Antoine hospital, Paris, France. They were collected during September 2022. The 348 stool was frozen at -80°C”
6. Overall analysis if AHL identification id depends on 12 samples in Figure 1 but only 2 IBD samples are taken for analysis. I could say clear dysbiosis of samples also, Why? How did u calculate sample size and suggest to add some more sample for IBD samples.
7. How did author concluded significance of AHL in the study from this statement ? “No synthase gene was found using blastn against raw reads. Regarding receptor 204 genes, no significant difference was found in terms of relative abundance in the meta-205 genome among non-IBD controls, CD, and UC patients”
Round 2
Reviewer 2 Report
All the comments are addressed